# Sustainability of Entrepreneurship: An Empirical Study on the Impact Path of Corporate Social Responsibility Based on Internal Control

**Xiao Guan [1], Chunli Yao [2] and Weimin Zhang [1],***

[1] School of Economics and Management, Beijing Forestry University, Beijing 100083, China; guanxiao@bjfu.edu.cn
[2] Audit Department, Beijing Normal University, Beijing 100875, China; 11112016014@bnu.edu.cn
* Correspondence: zwm@bjfu.edu.cn; Tel.: +86-10-62338012

**Abstract:** Effective internal control of enterprises can increase their social responsibility by improving financial performance, forming a sustainable cycle of enterprise development. This article uses relevant data from Chinese listed companies to explore the relationship between internal control, financial performance, and corporate social responsibility, as well as the differences in the impact of internal control on corporate social responsibility under the heterogeneity of property rights. We found that the three have a good promoting effect on each other; at the same time, financial performance plays a part in the media effect in corporate internal control and corporate social responsibility, and this effect is stronger in non-state-owned holding enterprises than in state-owned holding enterprises. This article suggests the following: (1) establish an internal control system for socially responsible enterprises and internalize corporate responsibility awareness; (2) strengthen the internal control and independent third-party supervision systems and form a joint internal and external supervision pattern; and (3) improve the top-level design of social responsibility and combine incentive and punishment measures. This study provides constructive suggestions for the sustainable development of Chinese listed companies and future research directions.

**Keywords:** internal control of enterprises; financial performance; CSR (corporate social responsibility)

## 1. Introduction

Good internal control can significantly improve the financial performance of a company and reduce the risks that it faces in production and operation [1]. The higher the level of internal control, the more likely it is that the company will achieve its business objectives [2]. Once a company has strong financial resources, it will be financially constrained to adopt social responsibility, and this will increase its enthusiasm for acting with social responsibility [3]. Good social responsibility performance helps enterprises establish a positive corporate image, improve the business environment, and establish strong relationships with stakeholders. It also helps enterprises obtain valuable external resources such as talent, funds, technology, etc. to add value to the enterprise and ensure its safe, stable, healthy, and sustainable development.

At present, the global economic growth rate is slowing down; at the same time, the pressure of the economic downturn has caused enterprises to excessively focus on their vested interests and to neglect their long-term interests. This has resulted in a serious lack of social responsibility [4,5]. Especially in recent years, some large enterprises at home and abroad have lost trust and collapsed due to a lack of social responsibility, and the lessons of internal control loopholes causing huge losses and even bankruptcy are shocking. The issue of the sustainable growth of enterprises is becoming increasingly prominent. Some practical and urgent issues facing enterprises are how to adapt to complex and dynamic environmental changes, how to enhance the enthusiasm of enterprises to fulfill their social

responsibilities, and how to achieve sustainable development for those enterprises. In this context, studying the influencing factors of corporate social responsibility fulfillment offers both theoretical and practical significance for expanding research on social responsibility and for promoting the construction of social responsibility.

The immoral behavior of enterprises is not only attributed to the weak sense of responsibility of management personnel but is also closely related to the operation of internal control and requires sufficient economic support. The lack of internal control leads to the failure to implement effective control and supervision in the business process, which in turn affects the assumption of social responsibility. The immoral behavior of enterprises is closely related to the operation of internal control [6–8]. Improving internal control, improving operational efficiency and effectiveness, valuing and fulfilling social responsibilities in the process of economic development, and striving to safeguard social public interests have become inexhaustible driving forces for enterprises to achieve sustainable development. In the process of achieving sustainable development in enterprises, how to strengthen the construction of internal control management systems, ensure the stable improvement of financial performance, strengthen the enthusiasm of enterprises to fulfill social responsibilities, and achieve a virtuous cycle of enterprise operation has become the focus of enterprise work.

State-owned holding enterprises, as directly invested or controlled by the government, have certain administrative and public welfare characteristics and should bear more social responsibilities compared to other enterprises. The management systems of non-state-owned holding enterprises are different from those of state-owned holding enterprises. The opportunistic motivation to assume social responsibility for fear of increasing operating costs is stronger, and its camouflage role is more obvious. Compared with state-owned holding enterprises, non-state-owned holding enterprises pay particular attention to economic performance; their economic motivation to undertake social responsibility is stronger than that of state-owned holding enterprises. Only when their financial performance is good can they have sufficient resources to ensure the fulfillment of their social responsibilities. Therefore, compared with state-owned holding enterprises, does the financial performance of non-state-owned holding enterprises have a more obvious media effect on the impact of internal control on corporate social responsibility?

To answer the above questions, this article uses data from Chinese listed companies from 2014 to 2018 as the research sample to explore the relationship between internal control and corporate social responsibility, and it deeply analyzes the path selection of financial performance as a mediator. This article aims to clarify the logical relationship between the three and provide useful suggestions for listed companies to strengthen internal control, achieve sustainable and good development, and assume more social responsibility based on research results.

This article may have the following marginal contributions: First, it fills the research gap in the existing literature by mainly focusing on the internal influencing factors of corporate social responsibility. From the perspective of internal and external stakeholders, this article explores the relationship between internal control and corporate social responsibility, studying financial performance and corporate social responsibility within the same framework, and exploring the concept that internal control not only has a direct impact on corporate social responsibility but that it can also have an indirect impact through the transmission mechanism of financial performance. Second, this article, starting from theoretical analysis and empirical research, analyzes the mechanism of the impact of internal control on corporate social responsibility and compares and analyzes the intermediary role of financial performance under different property rights in order to provide theoretical and practical references for managers to implement internal control, improve corporate financial performance, and actively assume social responsibility.

The remaining parts of this article are organized as follows: the second part introduces theoretical analysis and research hypotheses; the third part provides sample selection and data sources; variable definitions; and model construction; the fourth part discusses

descriptive statistics and correlation analysis between variables; the fifth part introduces regression analysis and a property rights grouping test; the sixth part tests the robustness of the model; and the seventh part introduces the research conclusions. In the final section, we will discuss these research findings and draw conclusions.

## 2. Theoretical Foundation and Research Hypotheses

### 2.1. Analysis of the Relationship between Internal Control and Corporate Social Responsibility

Corporate social responsibility is the product of social and economic development at a certain stage. From the perspective of contemporary management theory, it is the responsibility of social welfare or moral obligation that enterprises voluntarily or actively choose to undertake. In today's rapidly developing economy, many enterprises often overlook the negative impact of their own behavior on the overall development of society while they are pursuing economic benefits; this can ultimately threaten the normal operation of the enterprise itself. How to lead corporate governance with responsibility to achieve value co-creation between enterprises and society has increasingly become the focus of enterprise development.

From the development process of internal control, although internal control has undergone an evolutionary process from unconscious internal control to subjective intervention internal control and then to the adoption of policy regulation, its central idea of effectively ensuring the achievement of enterprise goals has not changed. Therefore, this article believes that "internal control is a process that is jointly followed and implemented by all levels of departments and employees to achieve corporate goals, ensure normal operation of the enterprise, and improve operational efficiency".

A sound corporate governance system cultivates correct values and social responsibility through the establishment of relevant organizational structures and corporate culture [9,10], enhances the willingness of enterprises to actively assume social responsibility in daily operations, is an important guarantee for fulfilling corporate social responsibility [11–13], and is also a key factor and institutional guarantee for maintaining its scientific development [14–16].

A good internal control management system is also conducive to cultivating risk awareness, strengthening enterprise risk management, and significantly reducing various risks faced by enterprises in fulfilling their social responsibilities [17–19]. It can be seen that the effective implementation of internal control is a fundamental prerequisite for corporate governance and that it can provide reasonable assurance for the achievement of control objectives.

Based on this, the following assumptions are proposed:

**H1.** *Internal control significantly positively affects corporate social responsibility.*

### 2.2. Analysis of the Relationship between Internal Control and Financial Performance

Internal control, as an organizational system, affects the sustainable competitiveness of enterprises from multiple dimensions, such as accounting information quality; scarce and difficult-to-imitate institutional resources; effective integration, construction, and re-configuration of non-institutional resources; and enhanced enterprise value [20]. Financial performance is the performance of enterprises investing in certain production factors, obtaining maximum use value and investment efficiency through operation and management, and the performance of profit, operation, and growth in a specific period of the enterprise operation process, reflecting the final business results of the enterprise. At present, in the evaluation of corporate financial performance, it is often analyzed from three dimensions: profitability, development ability, and shareholder profitability.

A good internal control system can effectively supervise and reduce managers' moral hazard, adverse selection, and other opportunistic behaviors [9], make up for the agency problems and shareholders' dissatisfaction caused by the unreasonable equity concentration ratio; alleviate the agency conflict between shareholders and managers; reduce the agency

cost; help enterprises enhance their solvency; reduce their financial risks; and improve their operational efficiency and future growth [21–23].

At the same time, effective internal control can reduce the cost of debt financing and the external risks of enterprises, and it can increase the sensitivity of investment expenditure and investment opportunities [24–26], restrain inefficient investment by enterprises, affect the risk premium of creditors, improve the efficiency of strategy implementation, increase the diversification value of enterprises, and promote the sustainable and stable development of enterprises by improving the function of strategy selection [27,28].

Based on this, the following assumptions are proposed:

**H2.** *Internal control significantly positively affects a company's financial performance.*

*2.3. Analysis of the Relationship between Corporate Social Responsibility and Financial Performance*

With the development of theory, discussions about the relationship between corporate social responsibility and financial performance have mostly focused on exploring how corporate social responsibility behavior affects financial performance, relatively neglecting the impact of financial performance on social responsibility [15].

Based on "resource theory", companies that achieve high financial performance will fulfill more social responsibilities. The idle resource hypothesis from this perspective suggests that, in order for enterprises to achieve environmental protection, energy conservation, and emission reduction, a large amount of funding needs to be invested in relevant equipment, technology, and human resources. A sustained and stable financial foundation provides routine cash flow guarantees for enterprises to fulfill their social responsibilities [29–31].

According to stakeholder theory, when a company achieves good financial performance, shareholders have a more positive attitude toward social responsibility investment compared to when the company's financial performance is poor. As a result, management will receive generous salary returns and will be more willing or conscious to pay attention to the rights and interests of stakeholders, increasing the company's social responsibility investment. After prioritizing the allocation of internal resources to meet daily economic activities, enterprises will only consider fulfilling social responsibility and other aspects. Therefore, financial performance is the main source of funds that encourages enterprises to undertake social responsibility, playing an important role in promoting corporate social responsibility.

Enterprises may do good deeds that are beneficial to society, or they may harm the interests of the stakeholders in their operations. Relevant scholars have proven that due to the widespread existence of "negative bias" among people, the value of enterprises is more influenced by their negative behavior than by their positive behavior. The good reputation gained by fulfilling social responsibility when a company encounters a crisis or a negative event can effectively alleviate the negative effects of the company or can alleviate legal sanctions, making it suffer fewer losses [16].

Therefore, the better the financial performance of a company, the more social attention and supervision it receives. In order to establish a good image, the company is willing to spend more funds on the practice of social responsibility [32,33]. Based on this, the following assumption is proposed:

**H3.** *Financial performance significantly positively affects corporate social responsibility.*

*2.4. The Mediating Role of Financial Performance*

An effective internal control system can control enterprise risks, safeguard the interests of various stakeholders, reduce transaction costs, promote the improvement of financial performance, ensure that enterprises have more material resources, and provide resource support for fulfilling social responsibilities. At the same time, a sound internal control system can improve the governance level and the resource utilization of enterprises, reduce

agency conflicts, enhance investment efficiency, and improve financial performance. Enterprises with good financial performance will attract more investors and maintain their long-term competitive advantage by winning a positive social reputation; thus, they have a stronger willingness to actively take on social responsibility.

From this, it can be seen that effective internal control can significantly improve the financial performance of enterprises, thereby improving the fulfillment of corporate social responsibility. As shown in Figure 1, financial performance plays a mediating role between internal control and corporate social responsibility.

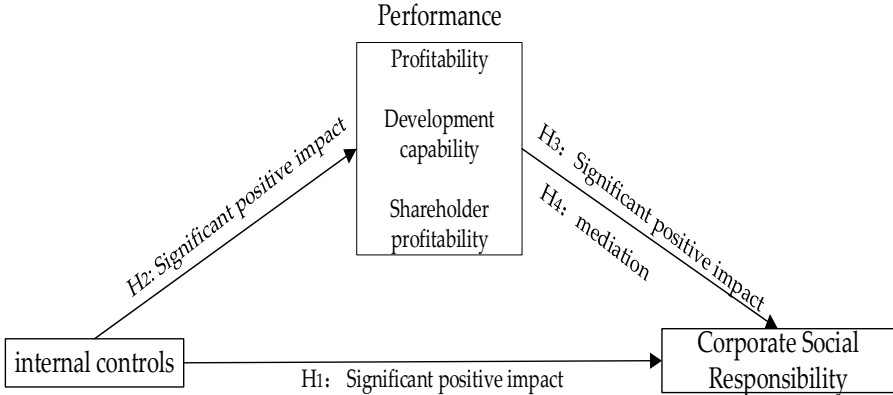

**Figure 1.** Transmission path of internal control, financial performance, and corporate social responsibility.

Based on the characteristics of the enterprises under China's special system, the nature of property rights has always been a hot topic of academic research. Considering the different nature of equity, it can be divided into state-owned holding enterprises and non-state-owned holding enterprises.

State-owned holding enterprises have the business goal of serving the public interest. While a state-owned enterprise is pursuing its own interests, it also needs to balance social and public interests. When there is a conflict between the two, the overall interests of society should be the main focus. This natural characteristic determines that state-owned holding enterprises need to bear more social responsibility. Non-state-owned holding enterprises have low requirements for their level of social responsibility practice; their main aim is to achieve profit value for the enterprise. Due to limitations in their own endowment, a lack of ability, or complex motives, they show insufficient social responsibility fulfillment.

Compared to non-state-owned holding enterprises, state-owned holding enterprises have a stronger enthusiasm for assuming social responsibility. Even in cases of poor financial performance, state-owned enterprises cannot change their goal of safeguarding the public interest. However, for non-state-owned holding enterprises, only when their financial performance is good can they have sufficient resources to ensure their practice of social responsibility. In the process of cooperation between state-owned holding enterprises and various stakeholders, they can gain positive social status through their natural attributes of ownership and "public welfare." The desire for a state-owned enterprise to achieve a good image and social reputation by improving its financial performance and by assuming social responsibility is not as strong as the desire for non-state-owned holding enterprises.

Therefore, non-state-owned holding enterprises are more inclined to obtain funds to undertake social responsibility by improving financial performance.

Based on this, this article proposes the following assumptions:

**H4.** *Financial performance plays a mediating role in the impact of internal control on corporate social responsibility.*

**H5.** *Compared with state-owned holding enterprises, the financial performance of non-state-owned holding enterprises has a stronger media effect on the impact of internal control on corporate social responsibility.*

## 3. Methodology

### 3.1. Sample Selection and Data Sources

This article uses annual data from the Guotai An Database, the Hexun Network, and the DiBo Internal Control Database. All A-share listed enterprises in major stock exchanges in China from 2014 to 2018, namely the Shanghai Stock Exchange and the Shenzhen Stock Exchange, were selected as the research samples, and preliminary screening of the data was conducted: (1) excluding financial and insurance companies; (2) excluding data related to ST, ST *, and PT listed companies; and (3) eliminating missing data. Considering that outliers will lead to instability in the research results, to avoid the impact of extreme values on the regression results, a 1%–9% truncation process was applied to all data, and 10,570 sample pieces of observation data were finally obtained. Excel, v. 2020, STATA, v. 17.0, and SPSS, v. 27, software were mainly used for the statistical analysis of data.

### 3.2. Variable Design

#### 3.2.1. The Dependent Variable

As an officially designated rating agency, Hexun.com is the first to provide corporate social responsibility ratings for Chinese listed companies. Its published scores for shareholder responsibility, public welfare donations, contribution value, and other subcategories of social responsibility have been recognized and used by many scholars in China. The scores for public welfare donations and contribution value do not include information on corporate profitability. Considering that fulfilling corporate social responsibility may have a negative impact on profitability, this article uses the social responsibility scores of the sub-items published by the Hexun Network indicators to measure the performance of corporate social responsibility.

#### 3.2.2. The Dependent Variable and Explanatory Variables

The internal control index, designed based on the five major internal control objectives released by DiBo Company, is the first indicator system to measure the internal control quality of all listed companies in China [34]. It covers five aspects: corporate strategy execution results, business returns, true and complete information disclosure, legal and compliant operation, and asset safety. Public information involving domestic and foreign official websites and industry authorities, as well as enterprise/industry risk information, internal control evaluation, internal control audit, internal control defects, violations of laws and regulations, related party transactions, mergers and acquisitions, reorganization, litigation, external audit, laws and regulations, internal control dynamics, and other relevant data and information that affect and reflect internal control and risk management, can fully reflect the actual situation of internal control in China. Therefore, this article uses the Internal Control Index to measure the quality of internal control.

#### 3.2.3. Mediating Variables and Control Variables

To avoid the limitations of evaluating financial performance from a single dimension, this article draws on relevant research [35–37], as shown in Table 1. This article studies total asset net profit margin, net asset net profit margin, earnings per share, and operating revenue growth rate from three dimensions: profitability, shareholder profitability, and development ability. Principal component analysis is used to obtain comprehensive evaluation indicators of enterprise performance. After principal component analysis, as shown in Table 2, two principal components were extracted, with a KMO value of 0.625. The Bartlett test was significant at the 1% level, and the cumulative contribution rate of the extracted principal components was 77%. This indicates that this method can be used to analyze the selected sample data.

**Table 1.** Definition of enterprise financial performance indicators.

| Primary Indicators | Secondary Indicators | Indicator Calculation and Explanation | Indicator Symbol |
|---|---|---|---|
| Profitability | Net profit margin on total assets | Net profit/average total assets | $U_1$ |
| | Net asset profit margin | Net profit/average net assets | $U_2$ |
| Shareholder profitability | Earnings per share | Net profit/total number of shares at the end of the period | $U_3$ |
| Development capability | Operating revenue growth rate | (Current year's operating revenue—previous year's operating revenue)/previous year's operating revenue | $U_4$ |

**Table 2.** Principal component analysis results.

| | Eigenvalue | Variance Contribution Rate (%) | Cumulative Variance Contribution Rate (%) |
|---|---|---|---|
| First principal component | 2.079 | 51.958% | 51.958% |
| Second principal component | 1.001 | 25.042% | 77.000% |
| KMO value | | 0.625 | |
| Bartlett test chi-square value | | | 11,664.842 *** |

Note: *** represents significance at the 1% level.

According to the variance contribution rate of the principal components, the comprehensive factor score of enterprise performance was calculated by weighting. The calculation formula for the comprehensive factor score is:

$$F = 0.6748\,X_{1'} + 0.3252\,X_{2'}$$

where $X_{i'}$ (i = 1, 2) is the standardized data of $X_i$ (i = 1, 2).

In order to control the impact of other factors on corporate social responsibility, the control variables selected in this paper include enterprise size, financial leverage, equity concentration ratio, growth, and the nature of property rights. In addition, industry and year dummy variables were added to control for industry-fixed effects and year-fixed effects. The specific variable definitions are shown in Table 3.

**Table 3.** Variable definition table.

| Variable Type | Variable | Symbol | Definition Description |
|---|---|---|---|
| Explained variable | Corporate social responsibility | Csr | Social responsibility rating scores of listed companies released by Hexun.com. |
| Explanatory variable | Internal controls | Ic | DiBo—Internal Control Index of Chinese Listed Companies. |
| Mediating variable | Enterprise performance | F | Comprehensive variables obtained from principal component analysis of $U_1$–$U_4$ variables. |
| Control variable | Enterprise size | Size | Natural logarithm of total assets at the end of the period. |
| | Leverage | Lev | Current asset-liability ratio. |
| | Ownership concentration | Cr1 | Shareholding ratio of the largest shareholder. |
| | Growth | Growth | Tobin Q value. |
| | Property nature | State | Virtual variable, 1 when belonging to a state-owned holding enterprise, otherwise 0. |
| | Industry | Ind | Virtual variable, set according to the secondary industry of manufacturing in the 2012 version. |
| | Time | Year | Virtual variable belongs to a certain year, take 1, otherwise take 0. |

*3.3. Model Construction*

This paper uses the hierarchical regression method [38,39], the Sobel method, and the Bootstrap method to test the media effect.

To test the impact of internal control on corporate social responsibility (Hypothesis 1), model (1) was established:

$$\text{Csr}_{i,t} = \alpha_0 + \alpha_1 \text{Ic}_{i,t} + \alpha_2 \text{Size}_{i,t} + \alpha_3 \text{Lev}_{i,t} + \alpha_4 \text{Cr1}_{i,t} + \alpha_5 \text{Growth}_{i,t} + \alpha_6 \text{State}_{i,t} + \Sigma \text{Ind} + \Sigma \text{Year} + \varepsilon_{i,t}. \tag{1}$$

To test the impact of internal control on financial performance (Hypothesis 2), model (2) was established:

$$F_{i,t} = \beta_0 + \beta_1 \text{Ic}_{i,t} + \beta_2 \text{Size}_{i,t} + \beta_3 \text{Lev}_{i,t} + \beta_4 \text{Cr1}_{i,t} + \beta_5 \text{Growth}_{i,t} + \beta_6 \text{State}_{i,t} + \Sigma \text{Ind} + \Sigma \text{year} + \iota_{i,t}. \tag{2}$$

To test the impact of financial performance on corporate social responsibility (Hypothesis 3) and the mediating effect of financial performance (Hypothesis 4), model (3) was established:

$$\text{Csr}_{i,t} = \lambda_0 + \lambda_1 \text{Ic}_{i,t} + \lambda_2 F_{i,t} + \lambda_3 \text{Size}_{i,t} + \lambda_4 \text{Lev}_{i,t} + \lambda_5 \text{Cr1}_{i,t} + \lambda_6 \text{Growth}_{i,t} + \lambda_7 \text{State}_{i,t} + \Sigma \text{Ind} + \Sigma \text{Year} + \mu_{i,t}. \tag{3}$$

To verify the above assumptions, this paper determines the econometric model to be selected through the F-test and Hausman test, as shown in Table 4. The running results of STATA all indicate rejection of the original hypothesis; therefore, this article chooses to use a fixed-effects model.

**Table 4.** Panel data model estimation.

| Inspection Items | Model 1 | Model 2 | Model 3 | Conclusion | Best Model |
|---|---|---|---|---|---|
| F-test | 210.41 *** (0.00) | 108.26 *** (0.00) | 246.09 *** (0.00) | Reject the original hypothesis. | Fixed-effects model |
| Hausman test | 687.52 *** (0.00) | 272.87 *** (0.00) | 581.08 *** (0.00) | Reject the original hypothesis. | |

Note: *** represents significant at the 1% level.

## 4. Correlation Analysis, Regression Analysis, and Property Rights Grouping Test

### 4.1. Correlation Analysis

The correlation analysis is shown in Table 5. The correlation coefficients between internal control, corporate social responsibility, and financial performance are 0.211 and 0.306, respectively, which are positively correlated at the 1% significance level.

**Table 5.** Pearson correlation test results.

| Variable | Csr | Ic | F | Size | Lev | Cr1 | State | Growth |
|---|---|---|---|---|---|---|---|---|
| Csr | 1.000 | | | | | | | |
| Ic | 0.211 *** | 1.000 | | | | | | |
| F | 0.445 *** | 0.306 *** | 1.000 | | | | | |
| Size | 0.198 *** | 0.033 *** | 0.062 *** | 1.000 | | | | |
| Lev | 0.042 *** | 0.138 *** | 0.183 *** | 0.552 *** | 1.000 | | | |
| Cr1 | 0.104 *** | 0.051 *** | 0.113 *** | 0.159 *** | 0.058 *** | 1.000 | | |
| Growth | 0.061 *** | 0.121 *** | 0.216 *** | 0.024 ** | 0.018 * | 0.052 *** | 1.000 | |
| State | 0.073 *** | 0.033 *** | 0.098 *** | 0.348 *** | 0.271 *** | 0.209 *** | 0.134 *** | 1.000 |

Note: *, **, and *** indicate significance at the 10%, 5%, and 1% levels, respectively.

Preliminary validation of Hypotheses 1 and 2. The correlation coefficient between financial performance and corporate social responsibility is 0.445, showing a positive correlation at the 1% significance level.

Preliminary validation of Hypothesis 3. Enterprise scale, equity concentration ratio, growth, and property right nature all significantly improved the implementation of corporate social responsibility at the level of 1%, and financial leverage negatively affected corporate social responsibility at the level of 1%. In addition, the correlation coefficients of all variables are less than 0.6, and most variables are less than 0.5, so it is preliminarily determined that there is no serious multicollinearity between the variables.

*4.2. Regression Analysis*

The regression results of the entire sample are shown in Table 6. First, the VIF of the model is less than two, indicating that there is no multicollinearity between the variables of the whole sample. Secondly, the F values of the model are all significant at the 1% level, and the adjusted R2 values are 0.2627, 0.2525, and 0.3714, respectively, indicating that the model as a whole has economic significance and good goodness-of-fit through the significance test.

**Table 6.** Full sample regression results.

| Variable | Model (1) Dependent Variable: Csr | VIF | Model (2) Dependent Variable: F | VIF | Model (3) Dependent Variable: Csr | VIF |
|---|---|---|---|---|---|---|
| Constant | −43.99957 *** (−15.23) | | −2.02015 *** (−22.64) | | −18.88818 *** (−6.91) | |
| Ic | 0.01488 *** (17.23) | 1.07 | 0.00071 *** (26.47) | 1.07 | 0.00605 *** (7.41) | 1.14 |
| F | | | | | 12.43049 *** (42.58) | 1.35 |
| Size | 2.60831 *** (22.21) | 1.80 | 0.07721 *** (21.28) | 1.80 | 1.64859 *** (14.89) | 1.88 |
| Lev | −12.85556 *** (−18.18) | 1.74 | −0.62006 *** (−28.38) | 1.74 | −5.14787 *** (−7.60) | 1.87 |
| Cr1 | 0.05160 *** (6.40) | 1.14 | 0.00306 *** (12.29) | 1.14 | 0.01353 * (1.80) | 1.16 |
| Growth | 1.17607 *** (4.05) | 1.09 | 0.18201 *** (20.28) | 1.09 | −1.08642 *** (−3.97) | 1.13 |
| State | 0.52317 ** (2.03) | 1.43 | −0.051309 *** (−6.44) | 1.43 | 1.16098 *** (4.87) | 1.44 |
| Ind | control | | control | | control | |
| Year | control | | control | | control | |
| N | 10570 | | 10570 | | 10570 | |
| Adj R2 | 0.2627 | | 0.2525 | | 0.3714 | |
| F | 45.84 *** | | 43.51 *** | | 74.46 *** | |

Note: parentheses represent *t*-values, with *, **, and *** indicating significant values at the 10%, 5%, and 1% levels, respectively.

In model (1), internal control is positively correlated with corporate social responsibility at a significance level of 1%, with a regression coefficient of 0.01488, verifying $H_1$. In model (2), internal control is positively correlated with financial performance at a significance level of 1%, with a regression coefficient of 0.00071, supporting $H_2$. In model (3), financial performance is positively correlated with corporate social responsibility at a significance level of 1%, and the regression coefficient between the two is 12.43049, verifying $H_3$. After introducing the intermediary variable financial performance in model (1), the two are still positively significant at the 1% level, but the regression coefficient decreases from 0.01488 to 0.00605 and the adjusted R2 increases from 0.2627 to 0.3714. The model has a better fitting effect, indicating that financial performance plays a partial mediating role in the relationship between internal control and corporate social responsibility, verifying $H_4$.

After using the stepwise regression method to verify the media effect, this paper uses the Sobel and Bootstrap tests to determine the media effect and the effect proportion of financial performance again.

The Sobel test results and the Bootstrap test results are shown in Tables 7 and 8, respectively, and the test results are consistent with the full sample regression results.

**Table 7.** Sobel test results for all samples.

|  | Effect Value | Standard Error | Z Statistic | *p* Value |
|---|---|---|---|---|
| Indirect effect | 0.00883 | 0.0004 | 22.4789 | 0.0000 |
| Direct effect | 0.00605 | 0.0008 | 7.41264 | 0.0000 |
| Total effect | 0.01488 | 0.0008 | 17.2327 | 0.0000 |
|  | The proportion of media effect = 59.341% | | | |

**Table 8.** Bootstrap test results for all samples.

|  | Effect Value | Bootstrap SE | 95% Confidence Interval |
|---|---|---|---|
| Indirect effect | 0.00883 | 0.00056 | (0.00766, 0.00989) |
| Direct effect | 0.00605 | 0.00086 | (0.00443, 0.00778) |
| Total effect | 0.01488 | 0.00100 | (0.01291, 0.01685) |
|  | The proportion of media effect = 59.341% | | |

Note: Bootstrap = 5000.

### 4.3. Property Rights Grouping Inspection

To further verify the impact of property rights on the relationship between internal control and corporate social responsibility, this article divides the entire sample into state-owned holding enterprises and non-state-owned holding enterprises for group testing.

From models (1), (2), and (3) in Tables 9 and 10, it can be seen that in state-owned and non-state-owned holding enterprises, internal control is positively correlated with corporate social responsibility and financial performance at a significance level of 1%; there is a positive correlation between financial performance and corporate social responsibility at the 1% significance level. After adding financial performance, the regression coefficients of internal control and corporate social responsibility in state-owned and non-state-owned holding enterprises decrease from 0.01355 and 0.01554 to 0.00611 and 0.00574, respectively, and the adjusted R2 increases from 0.3192 and 0.2362 to 0.3927 and 0.3790, respectively. The model-fitting effect is better, explaining that the financial performance of state-owned and non-state-owned holding enterprises plays a partial mediating role in internal control and corporate social responsibility.

**Table 9.** Regression analysis of state-owned holding enterprises.

| Variable | Model (1) Dependent Variable: Csr | VIF | Model (2) Dependent Variable: F | VIF | Model (3) Dependent Variable: Csr | VIF |
|---|---|---|---|---|---|---|
| Constant | −53.17728 *** (−10.68) |  | −2.11954 *** (−16.05) |  | −26.88507 *** (−5.53) |  |
| Ic | 0.01355 *** (9.12) | 1.10 | 0.00060 *** (15.38) | 1.10 | 0.00611 *** (4.21) | 1.17 |
| F |  |  |  |  | 12.40467 *** (21.21) | 1.43 |
| Size | 2.93611 *** (13.85) | 1.81 | 0.08046 *** (14.31) | 1.81 | 1.93800 *** (9.42) | 1.91 |
| Lev | −15.77923 *** (−11.69) | 1.76 | −0.64741 *** (−18.08) | 1.76 | −7.74832 *** (−5.83) | 1.92 |
| Cr1 | 0.018526 (1.22) | 1.24 | 0.00152 *** (3.75) | 1.24 | −0.00028 (−0.02) | 1.25 |
| Growth | 1.70852 *** (2.86) | 1.08 | 0.19839 *** (12.54) | 1.08 | −0.75245 (−1.31) | 1.13 |
| Ind | control |  | control |  | control |  |
| Year | control |  | control |  | control |  |
| N | 3779 |  | 3779 |  | 3779 |  |
| AdjR2 | 0.3192 |  | 0.2868 |  | 0.3927 |  |
| F | 24.61 *** |  | 21.26 *** |  | 33.15 *** |  |

Note: parentheses represent *t*-values, with *** indicating significant values at the 1% levels, respectively.

**Table 10.** Regression analysis of non-state-owned holding enterprises.

| Variable | Model (1) Dependent Variable: Csr | VIF | Model (2) Dependent Variable: F | VIF | Model (3) Dependent Variable: Csr | VIF |
|---|---|---|---|---|---|---|
| Constant | −34.78410 *** (−9.55) | | −2.12534 *** (−16.95) | | −8.07310 ** (−2.41) | |
| Ic | 0.01554 *** (14.82) | 1.08 | 0.00078 *** (21.49) | 1.08 | 0.00574 *** (5.92) | 1.16 |
| F | | | | | 12.56785 *** (39.30) | 1.34 |
| Size | 2.42865 *** (17.01) | 1.60 | 0.08411 *** (17.11) | 1.60 | 1.37161 *** (10.43) | 1.67 |
| Lev | −11.22069 *** (−13.87) | 1.61 | −0.60314 *** (−21.66) | 1.61 | −3.64051 *** (−4.82) | 1.72 |
| Cr1 | 0.06952 *** (7.42) | 1.08 | 0.00403 *** (12.51) | 1.08 | 0.01884 ** (2.21) | 1.10 |
| Growth | 1.11417 ** (3.52) | 1.08 | 0.16785 *** (15.42) | 1.08 | −0.99528 *** (−3.43) | 1.12 |
| Ind | control | | control | | control | |
| Year | control | | control | | control | |
| N | 6791 | | 6791 | | 6791 | |
| AdjR2 | 0.2362 | | 0.2438 | | 0.3790 | |
| F | 27.92 *** | | 29.07 *** | | 53.46 *** | |

Note: parentheses represent *t*-values, with, **, and *** indicating significant values at the 5%, and 1% levels, respectively.

The Sobel test results and the Bootstrap test results of property right grouping are shown in Tables 11 and 12, respectively, indicating that both state-owned holding enterprises and non-state-owned holding enterprises have a media effect; this is consistent with the full sample regression results.

**Table 11.** Sobel media effect test results for property rights grouping.

| Sobel Media Effect Test Results for State-Owned Holding Enterprises | | | |
|---|---|---|---|
| | Effect value | Standard error | Z Statistic | *p* Value |
| Indirect effect | 0.00744 | 0.00060 | 12.4528 | 0.00000 |
| Direct effect | 0.00611 | 0.00144 | 4.21422 | 0.00000 |
| Total effect | 0.00611 | 0.00148 | 9.16857 | 0.00000 |
| The proportion of the media effect = 54.908% | | | |
| Sobel Media Effect Test Results for Non-State-Owned Holding Enterprises | | | |
| | Effect value | Standard error | Z Statistic | *p* Value |
| Indirect effect | 0.00980 | 0.00052 | 18.85500 | 0.00000 |
| Direct effect | 0.00574 | 0.00098 | 5.92348 | 0.00000 |
| Total effect | 0.01554 | 0.00104 | 14.81660 | 0.00000 |
| The proportion of the media effect = 63.063% | | | |

Through comparative analysis, it can be seen that, compared with state-owned holding enterprises, financial performance accounts for a larger proportion of the media effect in non-state-owned holding enterprises, indicating that financial performance plays a stronger media effect in non-state-owned holding enterprises than in state-owned holding enterprises. This verifies $H_5$.

**Table 12.** Test results of property right grouping Bootstrap media effect.

| Test Results of the Bootstrap Media Effect for State-Owned Holding Enterprises | | | |
|---|---|---|---|
| | Effect value | Bootstrap SE | 95% confidence interval |
| Indirect effect | 0.00744 | 0.000780 | (0.00595, 0.00901) |
| Direct effect | 0.00611 | 0.001362 | (0.00340, 0.00873) |
| Total effect | 0.01355 | 0.001581 | (0.01045, 0.01665) |
| The proportion of the media effect = 54.908% | | | |
| Test Results of the Bootstrap Media Effect for Non-State-Owned Holding Enterprises | | | |
| | Effect value | Bootstrap SE | 95% confidence interval |
| Indirect effect | 0.00980 | 0.00064 | (0.00850, 0.01106) |
| Direct effect | 0.00574 | 0.00104 | (0.00376, 0.00783) |
| Total effect | 0.01554 | 0.00126 | (0.01308, 0.01800) |
| The proportion of the media effect = 63.063% | | | |

Note: Bootstrap = 5000.

## 5. Robustness Test

The variables studied in this article may have a causal relationship with each other, and due to the complexity of financial performance measurement, the indicator system described in the article is difficult to cover comprehensively. At the same time, there may be endogeneity issues caused by missing variables between internal control and financial performance. To ensure the credibility and accuracy of the research conclusions, this article selects the industry annual average of the internal control index and the lagged period of the internal control index as instrumental variables when verifying the endogeneity between corporate social responsibility and internal control, as well as financial performance and internal control. When verifying the endogeneity between corporate social responsibility and financial performance, financial performance with lagged periods 1 and 2 is chosen an instrumental variable. The test is performed using the 2SLS regression method.

The endogeneity test results of the entire sample and the property grouping are shown in Tables 13–15. The results show that, after using instrumental variables to control for endogeneity, there is still a positive correlation at the 1% level between internal control and corporate social responsibility, internal control and financial performance, and financial performance and corporate social responsibility across the entire sample, state-owned and non-state-owned holding enterprises; at the same time, the instrumental variables selected for the entire sample, state-owned holding enterprises, and non-state-owned holding enterprises do not have weak instrumental variable problems or over-identification problems. Therefore, the conclusions of this study are robust.

**Table 13.** Endogeneity test results for all samples.

| Variable | Model (1) Dependent Variable: Csr | Model (2) Dependent Variable: F | Model (3) Dependent Variable: Csr |
|---|---|---|---|
| Constant | −43.79732 *** (−13.29) | −2.665401 *** (−16.89) | −14.25315 (−2.37) |
| Ic | 0.0187952 *** (5.20) | 0.0005428 *** (3.14) | |
| F | | | 5.000258 *** (5.49) |
| Size | 2.142162 *** (14.20) | 0.1047159 *** (14.50) | 1.604506 *** (6.76) |
| Lev | −12.58767 *** (−14.05) | −0.5948336 *** (−13.87) | −4.651724 *** (−3.11) |
| Cr1 | 0.0639169 *** (6.52) | 0.0026468 *** (5.64) | 0.0227243 *** (1.49) |
| Growth | 2.342715 *** (6.06) | 0.1872134 *** (10.12) | −1.125075 * (−1.87) |

**Table 13.** *Cont.*

| Variable | Model (1)<br>Dependent Variable: Csr | Model (2)<br>Dependent Variable: F | Model (3)<br>Dependent Variable: Csr |
|---|---|---|---|
| State | −0.5230725 * | 0.0088658 | 1.370793 *** |
| | (−1.87) | (0.66) | (3.01) |
| Ind | control | control | control |
| Year | control | control | control |
| DWH inspection | 3.3234 | 0.000216 | 20.8425 |
| | (0.0683) | (0.9883) | (0.0000) |
| Weak identification test | 105.498 | 105.498 | 1041.63 |
| | (0.0000) | (0.0000) | (0.0000) |
| Over identified | 1.17156 | 0.431631 | 1.50667 |
| | (0.2791) | (0.5112) | (0.2196) |
| N | 3583 | 3583 | 3583 |
| Adj R2 | 0.3141 | 0.2505 | 0.3279 |
| F(Wald chil2) | 1513.43 *** | 1035.88 *** | 1432.94 *** |

Note: parentheses represent *t*-values, with *, and *** indicating significant values at the 10%, and 1% levels, respectively.

**Table 14.** Endogeneity test results of state-owned holding enterprises.

| Variable | Model (1)<br>Dependent Variable: Csr | Model (2)<br>Dependent Variable: F | Model (3)<br>Dependent |
|---|---|---|---|
| Constant | −41.76498 *** | −2.334555 * | −13.30316 ** |
| | (−9.99) | (−11.37) | (−2.47) |
| Ic | 0.0122165 *** | 0.0003751 *** | |
| | (2.96) | (1.85) | |
| F | | | 12.19409 *** |
| | | | (7.01) |
| Size | 2.095333 *** | 0.0919484 *** | 0.9723636 *** |
| | (10.66) | (9.52) | (4.87) |
| Lev | −11.88508 *** | −0.5773983 *** | −4.833649 *** |
| | (−9.47) | (−9.36) | (−3.79) |
| Cr1 | 0.0480133 *** | 0.0016873 ** | 0.0274208 ** |
| | (3.67) | (2.63) | (2.51) |
| Growth | 2.890089 *** | 0.176262 *** | 0.7387463 |
| | (5.48) | (6.81) | (1.47) |
| Ind | control | control | control |
| Year | control | control | control |
| DWH inspection | 0.261003 | 0.071943 | 9.35791 |
| | (0.6094) | (0.7885) | (0.0022) |
| Weak identification | 64.3276 | 64.3276 | 435.088 |
| | (0.0000) | (0.0000) | (0.0000) |
| Over identified | 0.573254 | 0.151836 | 2.2185 |
| | (0.4490) | (0.6968) | (0.1364) |
| N | 1446 | 1446 | 1446 |
| Adj R$^2$ | 0.4211 | 0.2775 | 0.6190 |
| F (Wald chil2) | 992.40 *** | 507.83 *** | 1617.51 *** |

Note: parentheses represent *t*-values, with *, **, and *** indicating significant values at the 10%, 5%, and 1% levels, respectively.

**Table 15.** Endogeneity test results of non-state-owned holding enterprises.

| Variable | Model (1)<br>Dependent Variable: Csr | Model (2)<br>Dependent Variable: F | Model (3)<br>Dependent Variable: Csr |
|---|---|---|---|
| Constant | −45.68472 *** | −3.003793 *** | −0.3759095 |
| | (−8.53) | (−12.01) | (−0.05) |
| Ic | 0.0248077 *** | 0.0007219 ** | |
| | (4.00) | (2.49) | |

**Table 15.** *Cont.*

| Variable | Model (1) Dependent Variable: Csr | Model (2) Dependent Variable: F | Model (3) Dependent Variable: Csr |
|---|---|---|---|
| F | | | 6.056224 *** |
| | | | (5.46) |
| Size | 2.257265 *** | 0.1234409 *** | 1.341478 *** |
| | (9.66) | (11.30) | (4.41) |
| Lev | −12.64481 *** | −0.5992617 *** | −3.819738 ** |
| | (−10.05) | (−10.19) | (−2.12) |
| Cr1 | 0.0670371 *** | 0.00311 *** | 0.0135737 |
| | (4.58) | (4.55) | (0.70) |
| Growth | 1.691011 | 0.1790916 *** | 0.1330899 |
| | (0.004) | (6.54) | (0.19) |
| Ind | control | control | control |
| Year | control | control | control |
| DWH inspection | 3.24456 | 0.155202 | 13.5802 |
| | (0.0717) | (0.6936) | (0.0002) |
| Weak identification | 39.7413 | 39.7413 | 547.946 |
| | (0.0000) | (0.0000) | (0.0000) |
| Over identified | 0.614796 | 0.64297 | 0.246875 |
| | (0.4330) | (0.4226) | (0.6193) |
| N | 2137 | 2137 | 2137 |
| Adj $R^2$ | 0.2745 | 0.2807 | 0.3046 |
| F (Wald chil2) | 766.57 *** | 724.38 *** | 704.86 *** |

Note: parentheses represent *t*-values, with **, and *** indicating significant values at the 5%, and 1% levels, respectively.

## 6. Discussions and Implications

This article breaks through the previous literature on the internal influencing factors of corporate social responsibility. Based on data from A-share listed companies on the main board from 2014 to 2018, internal control, financial performance, and corporate social responsibility are studied from the perspective of internal and external stakeholders within the same framework. The mechanism of the impact of internal control on corporate social responsibility is analyzed, and the intermediary role of financial performance under different property rights is compared and analyzed.

This article is based on data from A-share listed companies on the main board from 2014 to 2018, examining the path selection of internal control affecting corporate social responsibility through financial performance from the perspective of internal and external stakeholders and further exploring it by grouping according to the nature of property rights.

Research has shown the following: (1) that internal control significantly positively affects corporate social responsibility; (2) internal control significantly positively affects the financial performance of enterprises; (3) financial performance significantly positively affects corporate social responsibility; (4) financial performance plays a mediating role in the impact of internal control and corporate social responsibility; and (5) compared with state-owned holding enterprises, the financial performance of non-state-owned holding enterprises plays a stronger media effect in the impact of internal control on corporate social responsibility.

I hope to establish an internal control system for social responsibility, internalize corporate responsibility awareness, strengthen internal control and independent third-party supervision systems form a joint internal and external supervision pattern, strive to improve corporate financial performance, achieve self-economic benefits, improve the top-level design of social responsibility, combine incentive and punishment measures, deepen the concept of corporate social responsibility, implement corporate social responsibility behavior, and promote sustainable development of the enterprise.

### 7. Conclusions and Recommendation

The research on corporate social responsibility at home and abroad mainly focuses on the research on the internal influencing factors of corporate social responsibility, and some of the literature analyzes the relationship between internal control and corporate social responsibility, financial performance, and corporate social responsibility.

This paper breaks through the research shortcomings of the existing literature. From the perspective of internal and external stakeholders, internal control, financial performance, and corporate social responsibility are studied within the same framework. The mechanism of the impact of internal control on corporate social responsibility is analyzed, and the intermediary role of financial performance under different property rights is compared and analyzed.

Based on the results of this paper, the following suggestions are put forward, hoping to improve the concept of corporate social responsibility and implement the behavior of corporate social responsibility by improving internal control and improving financial performance:

First, establish an internal control system for social responsibility and internalize the awareness of corporate responsibility. Guide enterprises to attach importance to and actively fulfill social responsibilities, and when improving the internal control system, the awareness of social responsibility should be integrated into the framework of internal control and transmitted to every employee;

Second, strengthen the supervision system of internal control and independent third parties and form a common internal and external supervision pattern. Internal oversight is a guarantee for the achievement of internal control objectives, and at the same time, independent third-party supervision is required to avoid the supervision mechanism becoming a formality. In addition, make full use of the public opinion role of media organizations, mobilize the enthusiasm of society as a whole, and promote the faster development of social responsibility practice;

Third, improve the top-level design of social responsibility and combine incentives and punishments. At this stage, the enthusiasm of Chinese enterprises to fulfill their responsibilities is polarized, and even behaviors that harm society occur. It is necessary for government departments to create a legal environment for enterprises to actively fulfill their social responsibilities and should formulate different policies according to the classification of social responsibilities to ensure reasonable supervision of corporate behavior;

Fourth, strive to improve the financial performance of enterprises and achieve their own economic benefits. The key to corporate social responsibility is not only to have a sound internal control system but also to have strong financial resources. Compared with state-controlled enterprises with natural attributes of social responsibility, non-state-controlled enterprises should pay more attention to the impact of financial performance on social responsibility.

Research on corporate social responsibility at home and abroad mainly focuses on the internal influencing factors of corporate social responsibility. Some of the literature analyzes the relationship between internal control and corporate social responsibility, as well as financial performance and corporate social responsibility. No scholars have delved into the logical relationship between the three from the perspective of internal and external stakeholders and further explored the path choices of internal control affecting corporate social responsibility.

This article categorizes the nature of property rights in China and explores the differences in the mediating role of financial performance. However, due to the fact that China's state-owned and non-state-owned holdings are very different from those of other countries, they may not be universally applicable. Meanwhile, the research window of this article is established in a relatively stable social environment, and the research conclusions may not be applicable to turbulent social environments, so this article has certain limitations.

**Author Contributions:** Resources, C.Y.; Writing—original draft, X.G.; Writing—review & editing, W.Z. All authors have read and agreed to the published version of the manuscript.

**Funding:** This research received no external funding.

**Institutional Review Board Statement:** Not applicable.

**Informed Consent Statement:** Not applicable.

**Data Availability Statement:** Not applicable.

**Conflicts of Interest:** The authors declare no conflict of interest.

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
