# Peer review of "Sustainability of Entrepreneurship: An Empirical Study on the Impact Path of Corporate Social Responsibility Based on Internal Control"

_sustainability, doi:10.3390/su151612180_

Round 1

Reviewer 1 Report

Dear Authors and Editor,

After reading and analyzing the article, I write some observations:

- The title is objective and conveys the theme of the study carried out.

- The theme is interesting, social responsibility is increasingly gaining ground in business discussions, mainly due to demands from consumers and the community in general, which ends up impacting the reputation of companies.

- Abstract: I missed the presentation of the methodological aspects of the research.

- Introduction: well contextualized, it justifies the study, however I missed the objectives being very clear.

- Theoretical foundation and research hypotheses: I found this topic very interesting, objective, it presents the main contents related to the study. Uses updated references.

- Methodology: the section was divided into parts, it was well organized and presents well how the research was developed.

- Correlation analysis, regression analysis, and property rights grouping test - Robustness test: robust and well-presented statistical part; I only indicate to take care that there were tables that were left without the title on the same page, perhaps a smaller space between lines within the tables could be used.

- Discussions and Implications: I think that due to the robustness of the data, this item was weak; there is little dialogue with the theoretical framework.

- Conclusions: do not add much to the study; do not present limitations and suggestions for future work.

- Reference: There are references that do not mention the year of publication and are not single-spaced.

** I think that the Chinese reality with state and non-state holdings is very different from other countries, which is why it is important to discuss with other studies and/or point them out as suggestions for future work. Another fact is the data period from 2024 to 2018, since then we have lived in a different time, we have experienced a pandemic that shook all sectors, I think this should be pointed out and indicated as a limitation of the study and also indicated as a suggestion for future work.

I hope to have contributed with authors and with the journal.

Yours sincerely,

Author Response

Dear Reviewer,

        Thank you for your suggestion. All of your suggestions are very important and have important guilding significance for my future research work. Below are the modifications I have made based on your suggestions:

  • Thank you for your affirmation of the article title.
  • Thank you for your affirmation of the topic of the article.

  • At your suggestion, I have included a theoretical analysis of academic terminology in the second part of the article, Theoretical foundation and research hypotheses. In 2.1, the relevant concepts of corporate social responsibility, the development process of internal control and the definition of internal control are explained respectively. The concept and evaluation dimension of financial performance were added in 2.2. All are marked in red font.

  • Thank you for your suggestion. Based on your suggestion, this article aims to clarify the logical relationship between the three and provide useful suggestions for listed companies to strengthen internal control, achieve sustainable and good development, and assess more social responsibility based on research results.

  • Thank you for your recognition of the Theoretical foundation and research themes section.

  • Thank you for your recognition of the Methodology section.

  • Thank you for your suggestion. I have tried my best to reduce the spacing between the rows of the table. Some tables can be displayed on one page by adjusting and highlighted in red font. But some tables cannot display on one page due to the large number of rows in the data.

  • Thank you for your suggestion. This article has added relevant content on Discussions and Implications, further clarifying the theoretical contributions and research methods compared to previous studies, and highlighted them in red font.

  • Based on your suggestion, this article has added relevant research suggestions and limitations in the Conclusions section, and highlighted them in red font.

  • Thank you for your suggestion. I have added the year of the relevant references and set a single line spacing for all references, highlighted in red font.

  • Based on your suggestion, this article points out in the research conclusion that there are limitations in the nature of property rights in China and changes in the social environment, and highlights them in red font.

         Thank you again for your suggestion. I hope to learn more knowledge from you.

Yours sincerely,

Guan Xiao

Reviewer 2 Report

The article "Sustainability of Entrepreneurship: An Empirical Study of Corporate Social Responsibility" deserved better appreciation and attention from the reviewer.

This study provides an interesting insight into the matter.

it's a good job,

but must reinforce:

What is the relevance of the study?

What brought us back to the study?

What are the contributions?

Reinforce the purpose of study.

I miss an explanation of the model used.

Discussions and Implications is very short,

compare with other studies.

Develop the theoretical and practical implications

Conclusion, it's too short,

future lines of investigation are lacking.

I hope I was helpful,

No other subject,

A hug

Author Response

Dear Reviewer,

Thank you for your suggestion. Your suggestion has important guiding significance for my future research. Here are the modifications I made based on your suggestion:

-What is the relevance of the study?

Internal control, financial performance, and corporate social responsibility are the acts of corporate management. Internal control is a management method that is observed and implemented by departments at all levels and all employees to achieve enterprise goals, ensure the normal operation of enterprises, and improve operational efficiency. Financial performance is the maximum use value and investment efficiency obtained by enterprises through operation and management, and reflects the final business results of enterprises. Corporate social responsibility is the product of the development of the enterprise to a certain stage, and is the responsibility for social welfare or moral obligations that enterprises voluntarily or actively choose to undertake on the basis of strong financial resources. Therefore, this paper argues that the three are related.

-What brought us back to the study?

At present, the global economic growth rate is slowing down; at the same time, the pressure of economic downturn has caused enterprises to excessively focus on their vested interests and to neglect their long-term interests. This has resulted in a serious lack of social responsibility. Especially in recent years, some large enterprises at home and abroad have lost trust and collapsed due to lack of social responsibility, and the lessons of internal control loopholes causing huge losses and even bankruptcy are shocking.These realities led me to study the relationship between internal control, financial performance and corporate social responsibility。The article adds relevant explanations to the introduction and is marked in red lettering.

-What are the contributions?

By revealing the impact path of internal control on corporate social responsibility, this paper aims to provide constructive suggestions for the construction of internal control, the improvement of financial performance and the fulfillment of corporate social responsibility of listed enterprises in China, as well as future research directions.

-Reinforce the purpose of study.

Thank you for your suggestion. In the introduction section, the research purpose of the article has been added to further enhance its potential contributions, and highlighted in red font.

-I miss an explanation of the model used.

Thank you for your suggestions. This paper has added references related to Model selection. By drawing on previous research results, Sobel and Bootstrap Mesomeric effect test methods have been selected and marked in red font.

-Discussions and Implications is very short,

compare with other studies.

Thank you for your suggestion. In the Discussions and Implications section of this article, we have added relevant content such as research contributions, research foundations, and research content compared to previous studies, which have been highlighted in red font.

-Develop the theoretical and practical implications

Conclusion, it's too short,

future lines of investigation are lacking.

This article has added relevant concepts and development history of internal control, financial performance, and corporate social responsibility in the relevant theoretical section. In the introduction and related theoretical sections, it has added problems in corporate management in the current context, revised the title of Part 7, and added some suggestions to provide some reference for future management and implementation of corporate social responsibility behavior. Compared with previous research, Added the unique contributions of this article, while also pointing out the research shortcomings and potential future research directions. All the above are marked in red font.

Thank you again for your suggestion.

Xiao Guan

Reviewer 3 Report

The paper has several shortcomings:

  1. Inappropriate title: The title does not accurately reflect the objective of the paper.

  2. Lack of sample characterization: The paper fails to provide sufficient information about the sample used in the study.

  3. Insufficient detail on variable construction: The construction of the dependent and independent variables is not adequately explained.

  4. Absence of results discussion: There is no discussion of the results in the paper.

  5. Weak conclusions: The conclusions drawn in the paper are not strong or comprehensive.

Author Response

Dear Reviewer,

Thank you for your suggestion. Your suggestion has important guiding significance for my future scientific research. Below is my response to your suggestion:

-Inappropriate title: The title does not accurately reflect the objective of the paper.

Thank you for your suggestion. Based on your suggestion, I will change the title to: Sustainability of Entrepreneurship:  An Empirical Study on the Impact Path of Corporate Social Responsibility Based on Internal Control.

-Lack of sample characterization: The paper fails to provide sufficient information about the sample used in the study.

Thank you for your suggestion. Based on your suggestion, this article will In Sample selection and data sources, select A-share listed companies on the main board from 2014 to 2020 as research samples to All A-share listed enterprises in major stock exchanges in China from 2014 to 2018, named Shanghai Stock Exchange and Shenzhen Stock Exchange, were selected as the research samples.All the above are marked in red font. Information.

-Insufficient detail on variable construction: The construction of the dependent and independent variables is not adequately explained.

Thank you for your suggestion.

Based on your suggestion, this article will include 3.2.1 The dependent variable was previously modified to read:“As an officially designated rating agency, Hexun.com is the first to provide corporate social responsibility ratings for Chinese listed companies. Its published scores for shareholder responsibility, public welfare donations, contribution value, and other sub categories of social responsibility have been recognized and used by many scholars in China, and the scores for public welfare donations and contribution value do not include information on corporate profitability.”

Added in 3.2.2. The dependent variable Explanatory variables“Public information involving domestic and foreign official websites and industry authorities, as well as enterprise/industry risk information, internal control evaluation, internal control audit, internal control defects, violations of laws and regulations, Related party transaction, mergers and acquisitions, reorganization, litigation, external audit, laws and regulations, internal control dynamics and other relevant data and information that affect and reflect internal control and risk management can fully reflect the actual situation of internal control in China.”

Added in 3.2.3. Mediating variables and control variables“To avoid the limitations of evaluating financial performance from a single dimension, this article”

All the above are marked in red font.

-Absence of results discussion: There is no discussion of the results in the paper.

Thank you for your suggestion. Based on your suggestion, I have added the differences between this article and my predecessor's research, as well as the research basis and content, as well as potential contributions to this article through discussion of the results.All the above are marked in red font.

-Weak conclusions: The conclusions drawn in the paper are not strong or comprehensive.

In the section 7. Conclusion and Recommendation, relevant suggestions based on the research results, potential limitations of this study, and future research directions have been added. All the above are marked in red font.

Reviewer 4 Report

Thank you for the opportunity to read this article and review it. The article is written on a topical topic, but lacks clarity. So, I have a few comments:

1. In line 240, the authors indicate that "...1% - 99% of all data were truncated". Note - how much exactly?

2. How many companies were included in the analysis and how many were rejected? In other words, whether the research sample is representative.

3. Tab. 1 The "Operating Revenue Growth Rate" metric seems to have a bug in column #3. The minus sign in the formula should be a division sign.

4. The model in line 274 has two variables? Why so? According to Tab. 3 "Comprehensive variables ob-tained from principal component analysis of U1-U4 variables", while the formula lacks U1-U4 variables. No logic.

5. The formulas on page 8/19 are missing subscripts and superscripts. This interferes with the correct reading of the formulas and confuses the reader.

6. Tab. 3 reference is made to "DiBo · Internal Control Index of Chinese Listed Companies", but there is no specific database or website where this index can be consulted. Trying to reach this index, I found the article https://sciendo.com/es/article/10.2478/amns.2021.2.00185, which is generally similar to your article. Please indicate the references to the above-mentioned index.

7. You only use one index to measure the Management Level "DiBo · Internal Control Index of Chinese Listed Companies". There are no references to how the State tests social responsibility based on internal and external audits.

I hope that my comments will contribute to improving the quality of the article.

Author Response

Dear Reviewer,

Thank you for your suggestion. Your suggestion has important guiding significance for my future research. The following are the modifications I have made to the article based on your suggestion:

-In line 240, the authors indicate that "...1% - 99% of all data were truncated". Note - how much exactly?

A total of 10786 observation data samples were obtained, and 10570 observation data samples were finally obtained by censoring the data Outlier. Mark in red font in the 3.1 Sample selection and data sources section.

-How many companies were included in the analysis and how many were rejected? In other words, whether the research sample is representative.

In the regression, the values greater than 99% of the Quantile or less than 1% of the Quantile are replaced with missing values to avoid the impact of extreme values on the regression results. There are 216 Outlier in this paper, which may lead to untrue results. Therefore, before continuous variable regression, it is necessary to truncate the data to make the data more representative. Mark in red font in the 3.1 Sample selection and data sources section.

-Tab. 1 The "Operating Revenue Growth Rate" metric seems to have a bug in column #3. The minus sign in the formula should be a division sign.

Thank you for your suggestion. The formula has been modified to:(Current year's operating revenue - previous year's operating revenue)/previous year's operating revenue.

-The model in line 274 has two variables? Why so? According to Tab. 3 "Comprehensive variables ob-tained from principal component analysis of U1-U4 variables", while the formula lacks U1-U4 variables. No logic.

To avoid the limitations of a single dimension, this article adopts principal component analysis and uses dimensionality reduction techniques to replace the original multiple variables with a few comprehensive variables. By extracting components from the four dimensions U1-U4, two principal components are ultimately extracted. Therefore, only X1 and X2 principal components appear in the formula for calculating the comprehensive factor score. Based on the variance contribution rate of the principal components, the enterprise performance comprehensive factor score F is weighted and calculated, F represents the comprehensive variable of financial performance in this article.

-The formulas on page 8/19 are missing subscripts and superscripts. This interferes with the correct reading of the formulas and confuses the reader.

Thank you for your suggestion. The superscripts and subscripts of the formula have been modified and highlighted in red font in the original text.

-Tab. 3 reference is made to "DiBo · Internal Control Index of Chinese Listed Companies", but there is no specific database or website where this index can be consulted. Trying to reach this index, I found the article https://sciendo.com/es/article/10.2478/amns.2021.2.00185, which is generally similar to your article. Please indicate the references to the above-mentioned index.

Thank you for your suggestion. I have already cited 3.22 https://sciendo.com/es/article/10.2478/amns.2021.2.00185 This article should be highlighted in red font.

-You only use one index to measure the Management Level "DiBo · Internal Control Index of Chinese Listed Companies". There are no references to how the State tests social responsibility based on internal and external audits.

According to your suggestion, "public information related to domestic and foreign official websites and industry authorities, as well as enterprise/industry risk information, internal control evaluation, internal control audit, internal control defects, violations, Related party transaction, mergers and acquisitions, litigation, external audit, laws and regulations, internal control dynamics and other relevant data and information that affect and reflect internal control and risk management" are added in 3.2, By explaining that the DiBo database measures the efficiency and effectiveness of implementing internal control regulations through internal and external auditing information, we aim to study the impact of the effectiveness of internal control on social responsibility.

Reviewer 5 Report

The quality of this manuscript will be improved if authors consider some following points:

+ The period of research duration which is from 2014 to 2018 needs to update to nearest year.

+ All hypothesis is related to internal control. However, it is not mentioned in the research title.

+ There is no paragraph for theory of all academic terms in the paper.

+ In the Figure 1, readers could not see the H signals for all hypothesis.

+ The section of Discussion should be expanded more explanation and comparison with previous studies.

Need to improve language more professional.

Author Response

Dear Reviewer,

Thank you for your suggestion. Your suggestion has important guiding significance for my future research. Below are the modifications I have made based on your suggestion:

+ The period of research duration which is from 2014 to 2018 needs to update to nearest year.

Based on your suggestions, I have updated the data in the article to 2020, which is the most recent data disclosed by the database, and is marked in red letters.

+ All hypothesis is related to internal control. However, it is not mentioned in the research title.

At your suggestion, I have added internal controls to the title of this article to Sustainability of Entrepreneurship: An Empirical Study of Corporate Social Responsibility Based on Internal Control and marked in red letters.

+ There is no paragraph for theory of all academic terms in the paper.

At your suggestion, I have included a theoretical analysis of academic terminology in the second part of the article, Theoretical foundation and research hypotheses. In 2.1, the relevant concepts of corporate social responsibility, the development process of internal control and the definition of internal control are explained respectively. The concept and evaluation dimension of financial performance were added in 2.2. All are marked in red font.

+ In the Figure 1, readers could not see the H signals for all hypothesis.

In response to your suggestion, Figure 1 has been revised to indicate the correspondence between internal control, financial performance and corporate social responsibility, and to include a hypothetical H signal.

+ The section of Discussion should be expanded more explanation and comparison with previous studies.

This paper revises the title of Part VII, and adds research shortcomings in the existing literature, the main contributions of this paper, some suggestions, and some references for the future management and implementation of corporate social responsibility. All of the above are marked in red font.

Thank you again for your suggestion.

Xiao Guan

Round 2

Reviewer 2 Report

Good job.

Congratulations.

Author Response

Dear Reviewer,

Thank you for your suggestion. Your suggestion is of great significance for my future research. Below is the content that I have carefully revised based on your suggestion:

-What is the relevance of the study?

Internal control, financial performance, and corporate social responsibility are the acts of corporate management. Internal control is a management method that is observed and implemented by departments at all levels and all employees to achieve enterprise goals, ensure the normal operation of enterprises, and improve operational efficiency. Financial performance is the maximum use value and investment efficiency obtained by enterprises through operation and management, and reflects the final business results of enterprises. Corporate social responsibility is the product of the development of the enterprise to a certain stage, and is the responsibility for social welfare or moral obligations that enterprises voluntarily or actively choose to undertake on the basis of strong financial resources. Therefore, this paper argues that the three are related.

-What brought us back to the study?

At present, the global economic growth rate is slowing down; at the same time, the pressure of economic downturn has caused enterprises to excessively focus on their vested interests and to neglect their long-term interests. This has resulted in a serious lack of social responsibility. Especially in recent years, some large enterprises at home and abroad have lost trust and collapsed due to lack of social responsibility, and the lessons of internal control loopholes causing huge losses and even bankruptcy are shocking.These realities led me to study the relationship between internal control, financial performance and corporate social responsibility。The article adds relevant explanations to the introduction and is marked in red lettering.

-What are the contributions?贡献是什么?

By revealing the impact path of internal control on corporate social responsibility, this paper aims to provide constructive suggestions for the construction of internal control, the improvement of financial performance and the fulfillment of corporate social responsibility of listed enterprises in China, as well as future research directions.

-Reinforce the purpose of study.

Thank you for your suggestion. In the introduction section, the research purpose of the article has been added to further enhance its potential contributions, and highlighted in red font.

-I miss an explanation of the model used.

Thank you for your suggestions. This paper has added references related to Model selection. By drawing on previous research results, Sobel and Bootstrap Mesomeric effect test methods have been selected and marked in red font.

-Discussions and Implications is very short,

compare with other studies.

Thank you for your suggestion. In the Discussions and Implications section of this article, we have added relevant content such as research contributions, research foundations, and research content compared to previous studies, which have been highlighted in red font.

-Develop the theoretical and practical implications

Conclusion, it's too short,

future lines of investigation are lacking.

This article has added relevant concepts and development history of internal control, financial performance, and corporate social responsibility in the relevant theoretical section. In the introduction and related theoretical sections, it has added problems in corporate management in the current context, revised the title of Part 7, and added some suggestions to provide some reference for future management and implementation of corporate social responsibility behavior. Compared with previous research, Added the unique contributions of this article, while also pointing out the research shortcomings and potential future research directions. All the above are marked in red font.

Reviewer 3 Report

The authors have considered the suggestions made in the previous revision

Author Response

Dear Reviewer,

Thank you for your suggestion. Your suggestion is of great significance for my future research. Below is the content that I have carefully revised based on your suggestion:

-Inappropriate title: The title does not accurately reflect the objective of the paper.

Thank you for your suggestion. Based on your suggestion, I will change the title to: Sustainability of Entrepreneurship:  An Empirical Study on the Impact Path of Corporate Social Responsibility Based on Internal Control.

-Lack of sample characterization: The paper fails to provide sufficient information about the sample used in the study.

Thank you for your suggestion. Based on your suggestion, this article will In Sample selection and data sources, select A-share listed companies on the main board from 2014 to 2020 as research samples to All A-share listed enterprises in major stock exchanges in China from 2014 to 2018, named Shanghai Stock Exchange and Shenzhen Stock Exchange, were selected as the research samples.All the above are marked in red font. Information.

-Insufficient detail on variable construction: The construction of the dependent and independent variables is not adequately explained.

Thank you for your suggestion.

Based on your suggestion, this article will include 3.2.1 The dependent variable was previously modified to read:“As an officially designated rating agency, Hexun.com is the first to provide corporate social responsibility ratings for Chinese listed companies. Its published scores for shareholder responsibility, public welfare donations, contribution value, and other sub categories of social responsibility have been recognized and used by many scholars in China, and the scores for public welfare donations and contribution value do not include information on corporate profitability.”

Added in 3.2.2. The dependent variable Explanatory variables“Public information involving domestic and foreign official websites and industry authorities, as well as enterprise/industry risk information, internal control evaluation, internal control audit, internal control defects, violations of laws and regulations, Related party transaction, mergers and acquisitions, reorganization, litigation, external audit, laws and regulations, internal control dynamics and other relevant data and information that affect and reflect internal control and risk management can fully reflect the actual situation of internal control in China.”

Added in 3.2.3. Mediating variables and control variables“To avoid the limitations of evaluating financial performance from a single dimension, this article”

All the above are marked in red font.

-Absence of results discussion: There is no discussion of the results in the paper.

Thank you for your suggestion. Based on your suggestion, I have added the differences between this article and my predecessor's research, as well as the research basis and content, as well as potential contributions to this article through discussion of the results.All the above are marked in red font.

-Weak conclusions: The conclusions drawn in the paper are not strong or comprehensive.

In the section 7. Conclusion and Recommendation, relevant suggestions based on the research results, potential limitations of this study, and future research directions have been added. All the above are marked in red font.

Reviewer 4 Report

Good morning, Dear Authors

It seems that not all sources of literature have been included in the text. Please check again and add references. At least #36, 39 is missing.

It seems that the formula in line 302 is still useless and misleading. You have not responded to this comment. What is "F"?

The line indicates that "1% -99% truncation process was applied to all data". However, there is no information about what it was like before such a modification, and what the series looks like after the modification. Has the situation actually improved? Was the larger corporation removed or the smaller? Were companies on the brink of bankruptcy removed? What are these companies? What companies are left? From what industries?

Still no reference to preliminary data. No possibility to check or repeat the test.

The number of variables in formulas 1-3 does not match. Please double check that all variables are included.

At this point, the article still has little to do with sustainability.

Author Response

Dear Reviewer,

Thank you for your suggestion. Your suggestion has important reference significance for my future research. Your suggestion has important reference significance for my future research.

Firstly, regarding the suggestion you made last time, I have carefully revised it and submitted it in the system. However, for some unknown reason, it was not displayed,

Secondly, the following is the article that I have carefully revised based on your suggestions. Please criticize and correct it.

-It seems that not all sources of literature have been included in the text. Please check again and add references. At least #36, 39 is missing.

Thank you for your suggestion. In the original text, 3.2.3 Mediating variables and control variables have been added by number 36 in section 3.3 Reference 39 has been added to the Model construction and highlighted in red font.

-It seems that the formula in line 302 is still useless and misleading. You have not responded to this comment. What is "F"?

To avoid the limitations of a single dimension, this article adopts principal component analysis and uses dimensionality reduction techniques to replace the original multiple variables with a few comprehensive variables. By extracting components from the four dimensions U1-U4, two principal components are ultimately extracted. Therefore, only X1 and X2 principal components appear in the formula for calculating the comprehensive factor score. Based on the variance contribution rate of the principal components, the enterprise performance comprehensive factor score F is weighted and calculated, F represents the comprehensive variable of financial performance in this article.

-The line indicates that "1% -99% truncation process was applied to all data". However, there is no information about what it was like before such a modification, and what the series looks like after the modification. Has the situation actually improved? Was the larger corporation removed or the smaller? Were companies on the brink of bankruptcy removed? What are these companies? What companies are left? From what industries?

Still no reference to preliminary data. No possibility to check or repeat the test.

In the regression, the values greater than 99% of the Quantile or less than 1% of the Quantile are replaced with missing values to avoid the impact of extreme values on the regression results. There are 216 Outlier in this paper, which may lead to untrue results. Therefore, before continuous variable regression, it is necessary to truncate the data to make the data more representative. Mark in red font in the 3.1 Sample selection and data sources section.

-The number of variables in formulas 1-3 does not match. Please double check that all variables are included.

Thank you for your suggestion. As Formula 1 is designed to verify the relationship between corporate social responsibility and financial performance, it only includes the two main variables of corporate social responsibility and financial performance, as well as the related control variables. Formula 2 is to verify the relationship between internal control and financial performance, so it only includes the two main variables and related control variables of internal control and financial performance. Formula 3 is to verify the relationship between corporate social responsibility, internal control, and financial performance. Therefore, it includes three main variables and related control variables: corporate social responsibility, internal control, and financial performance.

-At this point, the article still has little to do with sustainability.

Thank you for your suggestions. Internal control, financial performance and corporate social responsibility are behaviors that affect the Going concern of an enterprise. Internal control is a management tool that is jointly observed and implemented by departments and all employees at all levels to achieve corporate goals, ensure normal operation of the enterprise, and improve operational efficiency. Financial performance refers to the maximum value and investment efficiency achieved by an enterprise through business management, reflecting the ultimate operating results of the enterprise and affecting its sustainable ability. Corporate social responsibility is the product of the economic development of an enterprise at a certain stage. It is the responsibility for social welfare or Moral obligation that the enterprise voluntarily or actively chooses to assume on the basis of having abundant financial resources, so as to establish a good positive image for the enterprise and help the sustainability of its long-term development. Therefore, this article believes that the three have a certain relationship with the sustainability of enterprise development.

Round 3

Reviewer 4 Report

The authors did not comply with the previous comments. There is also no answer file. There are still factual errors in the text.